# Conuping BSA-Seq and RNA-Seq Reveal the Molecular Pathway and Genes Associated with the Plant Height of Foxtail Millet (*Setaria italica*)

**DOI:** 10.3390/ijms231911824

**Published:** 2022-10-05

**Authors:** Yongbin Gao, Yuhao Yuan, Xiongying Zhang, Hui Song, Qinghua Yang, Pu Yang, Xiaoli Gao, Jinfeng Gao, Baili Feng

**Affiliations:** 1State Key Laboratory of Crop Stress Biology in Arid Areas, College of Agronomy, Northwest A & F University, Yangling 712100, China; 2Dexing Township Agro-Pastoral Comprehensive Service Center, Nyingchi 860700, China; 3Anyang Academy of Agricultural Sciences, Anyang 455099, China

**Keywords:** foxtail millet, plant height, BSA-Seq, RNA-Seq

## Abstract

Foxtail millet (*Setaria italica*) plays an important role in C4 crop research and agricultural development in arid areas due to its short growth period, drought tolerance, and barren tolerance. Exploration of the dwarfing mechanism and the dwarf genes of foxtail millet can provide a reference for dwarf breeding and dwarf research of other C4 crops. In this study, genetic analysis was performed using phenotypic data, candidate genes were screened by bulk segregant analysis sequencing (BSA-Seq); differentially expressed genes and metabolic pathways in different strains of high samples were analyzed by RNA sequencing (RNA-Seq). The association analysis of BSA-Seq and RNA-Seq further narrowed the candidate range. As a result, a total of three quantitative trait loci (QTLs) and nine candidate genes related to plant height were obtained on chromosomes I and IX. Based on the functional prediction of the candidate genes, we propose a hypothetical mechanism for the formation of millet dwarfing, in which, metabolism and MAPK signaling play important roles in the formation of foxtail millet plant height.

## 1. Introduction

The increasing frequency of extreme disasters caused by global warming, as well as the continuous depletion of cultivated land resources and degradation of cultivated land quality (due to the continuous expansion of industrialization), have posed significant challenges to the production of biological energy and food crops in today’s world [1,2]. As a result, most people acknowledge the importance of cultivating or exploring resistant and adaptable crops to address this global issue [3]. Foxtail millet originated in China and has a cultivation history of at least 16,000 years [4]. As an essential food and feed crop worldwide, its drought endurance, barren tolerance, and high-water usage efficiency, foxtail millet is widely farmed in arid and arid locations in India, China, and other parts of Asia, North Africa, and the Americas, and plays a vital role in the development of ecological agriculture [5]. Simultaneously, regarded as a strategic reserve crop for dealing with the future’s rapidly rising climate and arid ecological environment, it plays a key part in the current planting structure adjustment [6].

However, an excessively high plant height makes foxtail millet prone to lodging, reducing yield and quality, and severely limiting its development [7]. In most gramineous crops, improving plant type by reducing plant height is often considered to be an important technical means to improve yield and product quality [8,9]. Plant height is an important agronomic trait related to the high and stable yield of foxtail millet, and plays an important role in plant type, lodging resistance, and regional layout [10]. Although some QTLs or genes associated with plant height have been identified in foxtail millet, such as qPH1.1, q4, qph6, qph7 and qph9, qPH1-1, qPH1-2, qPH5-2, qPH5-3, qPH8, qPH9, SiD1, SiD2, and Seita.1G242300, their contributions to phenotypic variation are generally small or the mapping area is large [11,12,13,14,15]. Due to the lack of findings on efficient genes or QTLs, the application of dwarf lines in foxtail millet production is still rare [16].

Bulk segregation analysis (BSA) was first proposed by Michelmore et al. in 1991 [17]. It is a simple and efficient trait location method to construct a hybrid pool based on the extreme phenotype of a genetic segregation population and screen molecular markers associated with the phenotype through the frequency difference of mutation in the hybrid pool. BSA sequencing (BSA-Seq) can save a lot of time for population construction when locating QTLs. So, it is considered to be a fast and effective method for identifying functional genes or quantitative trait loci [18], and is widely used in the trait mapping of Arabidopsis, rice, maize, cucumber, and other species. Moreover, RNA-Seq was developed more than a decade ago and has since become a common tool in molecular biology [19]. Usually, there are more than one or several candidate intervals or sites obtained by BSA-Seq, and it is difficult to perform functional verification directly [20]. Thus, association analysis with RNA-Seq not only tends to further narrow the candidate range but also helps us achieve better status results through the gene’s structure and function analysis [21]. Li et al. [22] found the casein kinase gene BrCKL8, a key gene for forming Chinese cabbage leaf heads, by using BSA and RNA-Seq combined analysis, revealing the critical pathway of head formation in this crop. In a study on millet, Shareif Hammad Hussin et al. [23] mapped and analyzed the gene *SiMADS34* regulating panicle development by means of BSA-Seq and RNA-Seq.

In this study, to explore key genes related to plant height, an F_2_ population of 200 plants was obtained by hybridization between high straw (XYP1) and dwarf varieties (AYP1). Through the investigation and statistical analysis of plant height of 200 F_2_ populations, it was preliminarily speculated that the genes affecting plant height formation in this population were quantitative traits controlled by main effect genes. Based on it, we selected individuals with higher plant heights and shorter individuals to construct mixed pools of high stem bulk (F2H) and low stem bulk (F2L), respectively, for sequencing, and determined candidate regions and loci related to the plant height of millet by comparing allele frequencies. Functional genes and metabolic pathways that may cause plant height differences were analyzed by detecting differentially expressed genes of parents with large plant height differences before jointing. The association analysis between BSA-Seq and RNA-Seq was performed to further narrow the candidate region and identify eight candidate genes. It laid a foundation for future dwarf breeding, intensive production, and quality improvement of foxtail millet.

## 2. Results

### 2.1. Statistics and Analysis of Plant Height

Firstly, we found that the plant height of the parents showed significant differences during the jointing stage, and AYP1 had an obvious dwarfing phenomenon compared with XYP1 (Figure 1A,B). After maturity, we used IBM SPSS Statistics 25.0 to draw the plant height frequency distribution map of 200 populations and carried out a normal distribution test. The results showed that skewness = −0.654 < 1 and kurtosis = 0.419 < 1, which indicated that the plant height of 200 foxtail millet plants conformed to a normal distribution, so we speculated that the plant height of millet in this population was a quantitative trait controlled by the main effect gene (Figure 1C). Then, we sorted the plant height data from low to high in Excel and selected 30 plants of the dwarf and 30 plants of high-stem to form two extreme pools related to plant height (Figure 1D).

### 2.2. Bulked Segregant Analysis Sequencing

#### 2.2.1. Evaluation of BSA-Seq Quality

On NovaSeq 6000 sequencing platform, a total of 69,528,463,200 bp original data were obtained after sequencing the two parents and two extreme mixing pools using the PE150 sequencing strategy. By taking quality control measures, such as removing joints and low-quality reads, a total of 68,956,712,700 clean bases was obtained. The quality of sequencing data was evaluated, and the sequencing error rate, statistical Q20, Q30, and GC content were calculated as shown in Appendix A. A quality evaluation showed that the GC content of these reads was 46.31–46.94%, and the sequencing data had high quality (Q20 ≥ 95.82%, Q30 ≥ 89.79%), which could be used for further analysis.

#### 2.2.2. Mapping Analysis, Detection, and Annotation of SNP

After mapping the quality-controlled data with the Yugu 1 genome in the integrated database (ensembgenomes.org) as the reference genome, a total of 453,617,948 reads were obtained. The results showed that the mapping rate of the samples was 97.91–99.07%, the average sequencing depth of the parents was 10×, and the average sequencing depth of the mixed pool was 50 ×, indicating that the reference genome was uniformly random covered, and the mapping rate was high, which was conducive to the SNP screening and annotation next (Table 1). In order to obtain reliable SNPs, the UnifiedGenotyper model of GATK3.7 software was used for SNP detection, and VariantFiltration was used for filtering. After that, 1,918,974 SNPs were obtained after ANNOVAR annotation (Appendix A). Among them, 84,086 SNPs were located in exon regions, including 48,755 “non-synonymous” mutations, 981 “stop gain” mutations, and 161 “stop loss” mutations.

#### 2.2.3. Location of Candidate Regions and Screening of Genes

Based on the results of genotyping, 3,118,969 polymorphic markers were selected from homozygous SNP loci in offspring. After calculating the SNP index of 3,118,969 markers in two extreme progeny pools, the distribution of SNP index on the chromosome was mapped with 20 kb as window and 2000 bp as step length (Figure 2A,B). Next, we calculated the ∆ (SNP index) of the two pools and selected the 95% confidence level as the screening threshold after 1000 permutation tests. As a result, total of 9266 locus was finally determined, mainly distributed in 45,756,999 bp~58,883,703 bp of chromosome IX, 31,946,156 bp~42,023,774 bp of chromosome I, and 31,946,156 bp~38242209 bp and 20,018,104 bp~29,028,755 bp of chromosome VIII (Figure 2C). After non-synonymous mutation sites located in exons within the candidate region were further selected for analysis, a total of 329 sites of 128 genes were found to be involved (Appendix A).

### 2.3. Results of RNA-Seq

#### 2.3.1. Quality Control of Transcript Sequencing Data

After sequencing, a total of 300,673,474 raw reads were obtained from two parents (three biological replicates per parent), and 294,076,662 clean reads were obtained after removing adapter and low-quality reads. The sequencing results showed that the GC content of sequencing data was 53.57%~53.70%, and the sequencing quality was high (Q20 ≥ 98.35%, Q20 ≥ 95.26%), which could be further analyzed (Appendix A).

#### 2.3.2. Quality Assessment of RNA-Seq

After mapping the quality-controlled sequencing data of two parents to the reference genome of Yugu 1, the results showed that more than 94.72% of the clean reads were mapped on the reference genome (Table 2). Most of the mapped reads were concentrated in the exon region of the gene (Figure 3A,B), indicating that the sequencing results were consistent with RNA-Seq characteristics. Pearson correlation coefficient was used to evaluate the biological repeatability and correlation of the samples, indicating that the biological repeatability of the samples was reliable (Figure 3C). PCA dimensionality reduction analysis of six samples from two parents showed that all samples could be clustered into two groups with a large difference between the two parents but with little difference between biological replicates (Figure 3D). Therefore, the sample quality is reliable and can be analyzed later.

#### 2.3.3. Identification of Differentially Expressed Genes Related to Plant Height

Each sample’s gene expression level was examined using the “union” model of HTSeq software (v.0.12.3). When using FPKM > 1 as the criterion for gene expression, the range of gene expression in various samples was 21,732~22,566 (Appendix A). Firstly, the obtained gene expression matrix was standardized using the DESeq2 R package, and then the *p* value and error detection rate were determined using the negative binomial distribution model and FDR correction with Benjamini/Hochberg (BH), respectively. Using XYP1 as the control group and | log2 (FoldChange) | > 1 & q < 0.05 as the standard of gene differential expression, we obtained a total of 2817 DEGs, of which 1990 genes were downregulated and 827 genes were upregulated (Figure 3E).

#### 2.3.4. GO Classification and Enrichment Analysis of DEGs

The annotation results showed that all DEGs could be annotated into 2346 pathways with more than 1600 pathways downregulated, indicating that the occurrence of dwarfing was more likely to be caused by the regulation of downregulated genes. Then, GO enrichment analysis was performed on DEGs according to upregulation and downregulation, and classified according to biological process (BP), molecular function (MF), and cellular component (CC). After screening with the corrected *p*-value (FDR ≤ 0.05), it was found that the downregulated DEGs were mainly enriched in the molecular function pathway. Among them, the downregulated DEGs were enriched in 32 MF pathways and 3 BP pathways, but the upregulated DEGs were significantly enriched in the 3 biological pathways, including 5 pathways in the BP, 6 pathways in the CC, and 5 pathways in the MF (Figure 4A,B). According to the enrichment degree and hierarchical relationship of DEGs in each pathway, the downregulated DEGs are mainly enriched in several pathways, such as enzyme activity, macromolecular material binding, transmembrane transport, arsenate transport, and carbohydrate metabolism (Figure 4A). Upregulated DEGs are mainly related to DNA binding, ADP binding, nucleic acid binding, heterocyclic compound binding, and organic cyclic compound binding (Figure 4B).

Based on this, we infer that the downregulation of genes related to metabolism and ion transport may lead to the decrease in millet growth ability, while the upregulation of genes related to DNA building is mainly a mechanism to deal with this biologically harmful phenomenon.

#### 2.3.5. KEGG Enrichment Analysis of DEGs

To further explore the key pathway of dwarf formation in foxtail millet from the perspective of metabolism, we used TBtools to perform KEGG enrichment on all DEGs. With high-stem samples as controls, all DEGs were enriched in 198 pathways, of which 138 were downregulated and 60 were upregulated (Appendix A). Then, the pathways enriched by upregulated DEGs and downregulated DEGs were analyzed, respectively.

The majority of the downregulated DEGs were enriched in phenylpropanoid biosynthesis, metabolism, plant MAPK signaling pathway, biosynthesis of secondary metabolites, phenylalanine metabolism, phosphatidylinositol signaling system, selenocompound metabolism, diterpenoid biosynthesis, starch, and sucrose metabolism. Moreover, ABC transporters, plant–pathogen interaction, photosynthesis, taurine and hypotaurine metabolism, plant hormone signal transduction, glycolysis, carotenoid biosynthesis, mannose type O-glycan biosynthesis, stilbenoid biosynthesis, diarylheptanoid biosynthesis, and gingerol biosynthesis pathways were enriched by downregulated DEGs (Figure 5A). The majority of the upregulated DEGs were enriched by mismatch repair, DNA replication, isoquinoline alkaloids biosynthesis, nucleotide excision repair, phenylalanine metabolism, tyrosine metabolism, amino sugar, and nucleotide sugar metabolism. Moreover, the metabolism of amino acids, production of alkaloids, citric acid cycle, and so on, were also enriched by downregulated DEGs (Figure 5B).

The results of the KEGG enrichment analysis were basically consistent with the results of GO enrichment, which further illustrated that the pathway played an important role in the formation of millet plant height. At the same time, 11 downregulated DEGs were enriched in plant hormone transduction pathways, indicating that the occurrence of dwarfing in this study may also be related to hormone signal transduction.

### 2.4. Combined Analysis of BSA-Seq and RNA-Seq

#### 2.4.1. Identify the Key Genes Related to Plant Height

In order to further explore the key genes affecting the plant height of foxtail millet, the results obtained by BSA-Seq and RNA-Seq were combined to analyze. We compared the 128 candidate genes obtained by BSA-Seq with the 2817 DEGs obtained by RNA-Seq using the Wayne diagram to obtain common genes among them. As a result, a total of nine genes were mutated at multiple sites and differentially expressed in samples with different plant heights, mainly distributed on chromosome IX, and also distributed on chromosome I. Among them, six genes were downregulated and three genes were upregulated.

In particular, the results of BSA-Seq showed the candidate region on chromosome VIII, but the combined analysis with RNA-Seq showed that no candidate genes were located on chromosome VIII. Therefore, we analyzed the expressions of the mutated genes located in this candidate region in the RNA-Seq analysis (Appendix A), and the results showed that the genes were not expressed or the expression levels were not significantly different in the shoot tips of millet with different plant heights. Considering that the stem tip during the jointing stage of foxtail millet plays a key role in the formation of plant height, and the samples we used for RNA-Seq were the stem tip tissues of this period, we excluded this candidate region on chromosome VIII.

#### 2.4.2. Functional Analysis of Candidate Genes

To further understand the functions and pathways of these genes, we extracted the CDS sequences of the nine genes and annotated them on the Eggnog-mapper website to obtain their basic information, as shown in Table 3. Combined with the information in NCBI, Uniport, and the Google Academic Database, the function of the final candidate genes was further analyzed.

In chromosome IX, compared with the control, the expressions of SETIT034720mg, SETIT034843mg, SETIT035219mg, SETIT040190mg, and SETIT034904mg were downregulated, while SETIT033879mg was upregulated. Among them, SETIT034720mg mainly expresses in the nucleus and has the HOX domain, which mainly has the functions of regulating the DNA-binding transcription factor activity and RNA polymerase II-specific. SETIT034843mg and SETIT040190mg belong to the protein NRT1/PTR family 8.3 genes, which are genes related to the plant membrane, controlling the activities of transmembrane transporters; they mainly participate in the carbohydrate transmembrane transport in plants. SETIT035219mg was annotated as a hexose carrier protein HEX6, belonging to the main promoter superfamily and sugar transporter family. It has the function of regulating symporter activity and is mainly involved in the sugar transport in plants. SETIT034904mg belongs to the sterol desaturase family, mainly expressed in the endoplasmic reticulum and chloroplast. It has the function of catalytic activity and participates in the biosynthesis and metabolism of organic compounds such as lipids. SETIT033879 mg is a gene related to C-5 cytosine-specific DNA methylase, mainly expressed in the nucleus. It has the function of nucleotide binding and ATP binding and participates in the process of C-5 methylation of cytosine.

In chromosome I, SETIT017539mg was downregulated but SETIT019635mg and SETIT020559mg were upregulated compared to the control. Among them, SETIT019635mg belongs to the E2F/DP family, located in the nucleus. It is a transcription regulator gene involved in the RNA polymerase II regulation of transcription; the main function is regulating the binding of sequence-specific DNA. SETIT017539mg belongs to the peptidase A1 family, which is mainly involved in the proteolysis of the protein–peptide bond by regulating the activity of aspartate-type endopeptidase. SETIT020559 mg was annotated as a hypothetical protein in NCBI and the Uniport database, and no research on its function and homologous genes has been found so far.

### 2.5. Verification of RNA-Seq and Candidate Genes by qRT-PCR

To verify the reliability of RNA-Seq and the expressions of candidate genes, we extracted RNA from the shoot tips of high-stem and dwarf-stem parents and performed qRT-PCR on the six most likely DEGs. As a result, the relative expressions of the selected genes were consistent with the results of RNA-Seq, which proved the reliability of the results (Figure 6). Further analysis of the relative expression levels of these genes revealed that the expression levels of the selected genes in different samples showed significant or extremely significant differences. Compared with the high-stem samples, the expression of SETIT040190mg in the dwarf samples decreased most significantly (Figure 6A), while SETIT034904mg, SETIT035219mg, SETIT034843mg, and SETIT034720mg were all significantly downregulated (Figure 6B–E). However, the expression of SETIT019635mg showed a significant upward trend (Figure 6F). The results of qRT-PCR further showed that the genes related to nitrogen transport had the most significant changes during the dwarfing process, followed by the genes related to photosynthesis and photosynthetic products, which further verified that nitrogen transport and photosynthesis played crucial roles in the dwarfing process of foxtail millet.

## 3. Discussion

### 3.1. Excavation of Key Genes in Plant Height Contributes to the Promotion of Foxtail Millet

Foxtail millet is drought-resistant, barren-resistant, and is widely adaptable, which is an important crop for optimizing the regional layout of the planting industry and ensures the long-term development of grain crop production in arid and semi-arid regions [24]. The improvement of plant type not only has an important impact on increasing yield but also has an important impact on its industrial layout [25]. Identifying the molecular genetic basis of plant height is conducive to breeding new varieties of ideal plant types by molecular breeding methods, so that the ecological adaptability, water use efficiency, mechanical harvesting ability, and yield potential of crops can be comprehensively improved [26].

In the 1960s, the International Maize and Wheat Improvement Centre and the International Rice Research Institute led and launched the world-famous “Green Revolution” [27]. After that, people have made great success in the improvement of varieties characterized by dwarfing breeding, which has greatly promoted the yield of crops [28,29]. In recent years, breeders have also conducted a lot of research and exploration on foxtail millet dwarfing breeding. With the application of dwarf materials, such as Dwarf 88 and Dwarf 84133 in millet breeding, plant height was reduced to a certain extent, and the yield level of foxtail millet was significantly improved [30]. However, there were still many problems in the production of middle-dwarf varieties, especially dwarf varieties, such as premature senescence and poor yield traits [31]. Thus, there are few varieties of dwarf millet, which can be widely used in production. In this case, mining dwarf genes and analyzing the mechanism of plant height formation by modern means is not only conducive to the realization of high yields and super high yields of foxtail millet but also conducive to the further promotion of foxtail millet.

### 3.2. New Genes Controlling Plant Height on Chromosomes I and IX

With the progress of science and technology, millet genome sequencing has been completed [32]. In order to better promote the development of millet, researchers are paying more attention to the study of dwarf gene mining and its mechanisms on the basis of using dwarf resources and achieved some preliminary results [30]. Wang et al. [33] used the F_2_ mapping population of foxtail millet with large phenotypic differences as the material, selected SSR as the molecular marker, and used the complete interval mapping method to locate two QTLs that control plant height on chromosome 2 of foxtail millet. Zhao [34] obtained a semi-dominant dwarf gene SiDw1 on chromosome 9 of foxtail millet by map-based cloning. Moreover, researchers used high-density SNP markers to locate the plant height control sites on chromosomes 1, 4, 5, 6, 7, and 9 [11,12,26]. Although previous researchers have found many QTLs or genes on the foxtail millet’s plant height, these genes or QTLs are not really used in the innovation and promotion of millet dwarf germplasm resources.

In this study, two extreme mixing pools were constructed to explore the key genes controlling the plant heights of foxtail millet by using parents with significant differences in plant height as materials. Firstly, by BSA-Seq, five candidate regions (located on chromosomes 9, 8, 1, and 3) were obtained, and 329 candidate sites were located. Subsequently, the results of BSA-Seq were combined with RNA-Seq to further reduce the candidate regions to three, and nine genes with significantly different expression levels in parent materials with different plant heights were screened out for further analysis as key genes controlling plant height formation. Through a literature review and comparison of nucleic acid sequences and positions, we found that the candidate regions and gene loci we obtained were different from those mapped by previous studies, which were located at 36,542,096~37,591,717 bp on chromosome I, and 51,771,586~51,925,937 bp and 57,439,447~58,783,363 bp on chromosome IX. Therefore, we believe that further studies on these candidate regions and genes related to plant height are likely to play important roles in the “Green Revolution” of foxtail millet in the future.

### 3.3. Metabolism and the MAPK Signaling Pathway Play Important Roles in the Plant Height Formation of Foxtail Millet

Plant height is an important factor affecting plant type. Most are quantitative traits controlled by multiple genes, with complex genetic bases, and are often regulated by many physiological and molecular factors [35]. The dwarf omics data of related model plants show that dwarf-related genes are mainly related to metabolism and secondary metabolism regulation, cell division, hormone regulation, photosynthesis, transcription regulation, biosynthesis, cell apoptosis, signal transduction, and so on [36]. In this study, among the DEGs obtained from different plant height samples, 20 DEGs were concentrated in the phenylpropanoid biosynthetic pathway, 94 DEGs were concentrated in the metabolic pathway, 61 DEGs were concentrated in the secondary metabolite biosynthetic pathway, and 16 DEGS were concentrated in the MAPK signaling pathway, indicating that the metabolic process and signal transduction had important effects on plant height.

Phenylpropanoid metabolism is an important secondary metabolic pathway in plants, and its metabolites, such as lignin, pollen, anthocyanin, and organic acids, play important roles in regulating plant adaptive growth [37]. Studies have shown that lignin produced by phenylpropanoid metabolism has an important effect on plant height formation by controlling the formation of the cell wall [38]. Luo et al. found that 22 DEGs in samples with different plant heights were annotated into the lignin monomer synthesis pathway when studying the dwarf lines of Brassica napus with specific plant types through transcriptome [39]. Huang et al. studied the candidate regulators of secondary cell wall (SCW) formation in rice and found that some lignin biosynthesis genes were negatively regulated by the OsIDD2 [40]. After overexpression of OsIDD2 in plants, the expression of lignin synthesis genes decreased, resulting in plant dwarfing, increased leaf brittleness, and decreased lignin content [40]. Moreover, the significant effects of lignin and phenylpropanoid metabolism on crop plant height have also been confirmed in wheat, maize, and other crops in recent years [41,42,43].

Mitogen-activated protein kinases (MAPKs) are the serine/threonine protein kinases in eukaryotes; they are phosphorylated by MAPKK and become activated, and then transfer signals by phosphorylating the serine or threonine residue of the substrate protein [44,45]. Together with some other signaling molecules, they form the MAPK cascade signaling pathway, which affects the expression of specific genes by transferring signals into cells after receiving signals from external stimuli [46]. The diversity of MAPK substrates and their different spatiotemporal expressions endow the MAPK cascade reaction with a great ability to control various biological processes [47,48]. In Arabidopsis, MPK3/MPK6 cascades downstream of ER/ERLs, which play important roles in promoting local cell proliferation, and determine the inflorescence structure, organ shape, and size [48,49,50]. Studies by Gao et al. showed that the destruction of the MEKK1-MKK1/MKK2-MPK4 pathway could lead to the dwarfing of Arabidopsis [51]. At the same time, many studies have shown that MAPK-mediated auxin signal transduction and polar transport pathways have significant effects on the plant height of maize, Arabidopsis, and so on [52,53,54,55]. In our study, more than 16 DEGs were annotated to the MAPK signaling pathway, so we speculated that the MAPK signaling pathway plays an important role in signal transduction, affecting plant height during the growth and the development of foxtail millet.

Based on the above conclusions, we speculated that the decrease in millet plant height may be due to some changes in the upstream, resulting in changes in MAPK signal transduction, resulting in lignin synthesis and other metabolic effects, and ultimately affecting the plant height of foxtail millet.

### 3.4. Nitrate Transporter Gene and Hormone Transport Gene Were Key Genes Controlling Plant Height Formation in Foxtail Millet

In this study, we obtained nine candidate genes related to plant height by combining BSA-Seq with RNA-Seq, of which, six were downregulated and three were upregulated in dwarf samples compared with the control. By analyzing the function of these genes with the help of the literature and related websites, we found annotations in millet or near-source crops, except for one gene (SETIT020559mg). Among them, downregulated genes included a homologous box protein BEL1 homologous gene, a sugar transporter family gene, a sterol desaturase family gene, a peptidase A1 family gene, and two protein NRT1/PTR family 8.3 genes.

Homologous box genes are found in almost all eukaryotes and are involved in all aspects of growth and development regulation [56]. In Arabidopsis on cytokinin regulating ovule development by regulating *PIN1*, it was confirmed that the transcription factor BEL1 is necessary for the correct mode of auxin and cytokinin signaling pathways [58]. Studies on rice have shown that the Os05g0455200 gene plays a redundant role in the establishment and maintenance of meristem and stem elongation, and the mutant created by it has an early termination codon in the upstream of the BEL1 domain, resulting in significant internode shortening and plant dwarfing; the researchers also speculated that the phenomenon is related to the polar transport of growth hormone [57]. Combined with the phenomenon that a homeobox protein BEL1 homologous gene (SETIT034720mg) was significantly downregulated in dwarf foxtail millet in this study, we speculated that the homeobox protein BEL1 homologous gene in millet may affect the plant height of millet by regulating the synthesis and transport of the growth hormone.

Photosynthesis is a process in which green plants use photosynthetic pigments to convert carbon dioxide and water into the organic matter under visible light irradiation. It is the most basic material metabolism and energy metabolism in the biological world. Carotenoids act as auxiliary pigments, photo-protectants against photooxidation stress, and regulators of plant growth and development during photosynthesis [57]. Mutations in carotenoid biosynthesis genes that reduce carotenoid levels in plants and, thus, affect plant photosynthesis were demonstrated long ago [59,60]. In rice, mutations in HTD12 disrupt the function of the carotenoid isomerase, leading to reduced carotenoid levels and resulting in plant dwarfing, which proves that carotenoid biosynthesis regulates photosynthesis and plant structure [61]. In addition, the reduction of carotenoids caused plant dwarfing in Arabidopsis, and upland cotton; other crops have also been confirmed [62,63,64]. In this study, a gene (SETIT034904mg) with the catalytic activity of carotenoid containing fatty acid hydroxylase domain decreased in dwarf samples. We speculate that this gene may affect the photosynthesis of plants by regulating carotenoids, thus resulting in the dwarf phenomenon.

Carbohydrates produced by photosynthesis of plant source organs (mainly leaves) are transported to its various sink organs (roots, flowers, and seeds) to support the overall growth and development of plants [65], and assimilate, usually in the form of sugar, entering phloem as a symplast or apoplast [66,67]. Sugar transporters not only mediate the transport of sugar in the phloem but also mediate the realization of different functions in plant growth and development because of their different distribution sites [68,69]. AtSWEET4 (a sugar that will eventually be the exported transporter gene) in Arabidopsis mediates the axial transport of sugar during plant development, and the knockout of AtSWEET4 reduces plant biomass [70]. In Camellia sinensis, the sugar transporter CsSWEET17 can affect the photosynthetic efficiency of plants by passing sugar through the cell membrane, thereby transferring more sugar to developing tissues for cell growth and affecting the growth of vegetative organs [71]. In rice, the function missing of a monosaccharide transporter (MSTs) leads to the limitation of the monosaccharide transport, which not only reduces the supply of sugar in plants, but also limits the growth of roots, thus limiting the absorption of potassium ions and other substances from the outside, and ultimately affects the growth of vegetative organs [72]. Because an MFS domain-containing protein gene (SETIT035219mg) was downregulated in dwarf samples in this study, we speculated that the decrease of the MFS domain-containing protein gene expression in foxtail millet affected the monosaccharide transport, thus affecting the energy supply and other nutrient uptakes during its growth process, resulting in dwarfing. However, researchers still need to explore why the expression of the MFS domain-containing protein gene is downregulated.

Nitrogen metabolism is closely related to plant dwarfing [27,73]. The absorption of nitrate ions is the first step in the process of nitrate nitrogen utilization in plants. The nitrate transporter (NRT) plays an important role in absorbing and sensing external nitrate and signal regulation, so it is considered to be one of the key biological macromolecules in the plant’s nitrogen nutrition system [74]. Low-affinity nitrate transporters play a role when the concentration of exogenous nitrate nitrogen is high, mainly encoded by the NRT1/PTR family (NPF) [75,76,77]. Studies on plant NPF have found that it is not only a nitrate or peptide transporter, some NPF transporters can also transport different substrates, such as nitrate/auxin, nitrate/abscisic acid, nitrate/glucosinolate, or gibberellin/jasmonic acid [78,79]. In rice, OsNPF3.1 is a member of the NRT1/PTR family, which affects the plant height of rice by increasing the nitrogen use efficiency of rice [80]. Guo et al. found that the decrease of NRT1 and NRT2 family genes can lead to the decrease of nitrogen in plants and affect plant height [76]. In particular, we found that the expressions of two protein NRT1/PTR family 8.3 genes (SETIT034843mg and SETIT040190mg) in the candidate region of a plant‘s height were significantly reduced in the dwarf sample, and the resequencing results showed that there were multiple non-synonymous mutations in the exon region of this candidate gene. We speculate that it is very likely that the mutations of these two gene loci lead to the decrease of the expression of these two genes, which affects the nitrogen transport in millet and causes the decrease in plant height.

Moreover, a gene of C-5 cytosine-specific DNA methylase is also downregulated on chromosome IX, which we infer is regulated by upstream genes in plants [81]. On chromosome I, a gene belonging to the peptidase A1 family (SETIT017539mg) was also downregulated in dwarf samples, which mainly regulates aspartic endopeptidase activity during protein hydrolysis, which may be related to ion transport in plants [82]. In addition, we infer that the increased expression of another gene with the E2F/DP family winged-helix DNA-binding domain and a hypothetical protein gene in dwarf samples is a stress response mechanism of the crop itself to the abiotic stress that causes dwarfing.

Based on the above results, we propose the following hypothesis about the formation process of millet dwarfing. First, mutations in upstream genes, such as phenylpropanoid-related genes, affect the morphogenesis of cells, including cell walls. Then the affected cells or cell walls will inevitably weaken in signal transduction and material transport. Finally, affected by the transport capacity weakened, nitrogen transport, and transport of photosynthetic substances in plants were also weakened, affecting the development and metabolism of millet, and ultimately leading to the occurrence of dwarfing. In this process, MAPK signal transduction and metabolic pathways play important roles.

## 4. Materials and Methods

### 4.1. Technical Route

In this study, XYP1 and AYP1, and their F_2_ hybrid populations, were studied. BSA-Seq and RNA-Seq were combined to explore the key genes and formation mechanisms that control the formation of foxtail millet plant height (Figure 7).

### 4.2. Plant Materials

Two foxtail millet varieties, Xinong 8852 (XYP1) and An 15 (AYP1), were used for hybridization in this study. After harvest, they were further planted and harvested (of the F_2_ generations). In April 2021, parents and F_2_ populations were planted in the Crop Teaching Specimen Area of Northwest A & F University [83]. Plant spacing was 10 cm and row spacing was 40 cm. At the elongation stage, the parents and 200 plants were listed and numbered, the fresh leaves were cut and frozen in liquid nitrogen, respectively, and then placed in a −80 °C refrigerator for DNA extraction. At the same time, the shoot tips of 3 biological repeats per parent were frozen in liquid nitrogen and stored in a −80 °C refrigerator for RNA extraction. Normal water and fertilizer management were conducted throughout the whole growth period.

### 4.3. Investigation and Statistics of Plant Height

After maturity, the plant heights of parents and 200 plants of the F_2_ population were measured by a ruler. Each parent of the two parents selected 10 plants to measure their plant height, and each plant in the F_2_ population was measured. The height from ground to spike top at the maturity stage was taken as the plant height of foxtail millet. SPSS software was used to analyze the plant heights of 200 groups, and Excel software was used to select 30 strains of low plant heights and 30 strains of high plant heights to construct an extreme mixing pool.

### 4.4. BSA-Seq

#### 4.4.1. DNA Extraction, Library Construction, and Sequencing

According to the statistical analysis of the plant height data of millet in the field, 30 high-stem millets and 30 dwarf foxtail millets were selected to form two extreme groups. The DNA of parents and two extreme mixing pools were extracted from each sample via the CTAB method [84]. The purity of DNA (OD260/OD280) was detected by a Nanodrop instrument (Nano-Drop Technologies, Wilmington, NC, USA) and the integrity and concentration of DNA were identified by Qubit Fluorometer (Thermo Fisher Scientific, Waltham, MA, USA) and agarose gel electrophoresis. Then qualified samples were sent to Beijing Ovison Technology Co., Ltd. (Beijing, China) for database construction and high-throughput sequencing (NovaSeq 6000; Illumina, San Diego, CA, USA).

#### 4.4.2. Comparison with Reference Genomes, Detection, and Annotation of Single-Nucleotide Polymorphisms

The BWA software 0.7.17 (parameter: mem-t 4-k 32-M) was used to compare the effective sequencing data to the reference genome, and the initial alignment results of the BAM format were obtained. Then, SAMtools was used to sort the results (parameter: sort, rmdup). To obtain reliable SNPs, we used the Unified Genenotyper model of GATK3.7 software for SNP detection and VariantFiltration for filtering. Finally, ANNOVER software was used for SNP annotation.

#### 4.4.3. Identification of Candidate Regions and Genes

The SNP frequencies (SNP index) of the two extreme mixing pools were calculated SNP using parents as references. Among them, the SNP index that was completely identical to the parent was 0, and the SNP index that was completely different from the parent was 1. In order to reduce the impact of sequencing errors and alignment errors, we filtered the polymorphic loci of parents after calculating the SNP index and filtered out the loci with a SNP index that was less than 0.3 and a SNP depth less than 7 in both pools, as well as the loci with an index missing in any pool. The difference in allele frequency in two extreme mixing pools was calculated by Δ(SNP index) = SNP index (Bulk B) − SNP index (Bulk A). Finally, 1000 permutation tests were performed on the calculation results, and a 95% execution level was selected as the threshold to screen candidate intervals and candidate loci.

### 4.5. RNA-Seq

#### 4.5.1. RNA Extraction, Library Construction, and Sequencing

The total RNA was extracted from the shoot tips of the two parents using the RNeasy Plant Mini Kit (Qiagen, Duesseldorf, Germany) according to the instructions. NanoDrop (Nano-Drop Technologies, Wilmington, USA) and Agilent 2100 Bioanalyzer (Agilent Technologies, USA) was used to detect the purity of RNA and the lengths of RNA fragments; agarose gel electrophoresis was used to detect RNA integrity. The qualified samples were sent to Beijing Ovison Technology Co., Ltd., for library construction and sequencing (NovaSeq 6000; Illumina, San Diego, CA, USA).

#### 4.5.2. Sequencing Data Quality Control and Reads Mapping

Use Trimmomatic software (v0.33) to filter the original data obtained by sequencing, and reads with the ratio of joint, N (uncertain base) content greater than 10%, and low-quality base (Q ≤ 20) content greater than 50% were removed. We used STAR software and set the default parameters to map the quality control data to the reference genome.

#### 4.5.3. Gene Expression Level Analysis

To prove the repeatability of the experiment and the reliability of the experimental results, we used the square of Pearson’s correlation coefficient (R^2^) between samples to verify the reliability of the experiment. The principal component analysis (PCA) was used to test the correlation of samples. After obtaining the gene expression matrix, the gene expression level of each sample was analyzed by the union model of HTSeq software. We identified differentially expressed genes (DEGs) between parents using the DESeq2 R package [85].

#### 4.5.4. Functional Analysis of DEGs

Gene Ontology (GO, www.geneontology.org, accessed on 20 August 2022) is an international standardized gene functional classification system, which divides the functions of genes into three parts: cellular component (CC), molecular function (MF), and biological process (BP). By a GO enrichment analysis, we could obtain what the differentially expressed genes were mainly related to at CC, MF, and BP levels. The Kyoto Encyclopedia of Genes and Genomes (KEGG, www.kegg.jp, accessed on 20 August 2022) is a database related to gene action pathways, which not only includes metabolic pathways involved in gene locks but also comprehensively annotates enzymes that catalyze each step of such reactions. These two databases not only help us better understand the function of DEGs but also help us more effectively select research-related target genes for further analysis. After obtaining the GO and KEGG annotation files of the whole genome through EggNOG-mapper (eggnog-mapper.embl.de), TBtools was used for enrichment analysis [86].

### 4.6. Combined Analysis of BSA-Seq and RNA-Seq

We combined the candidate sites obtained by BSA-Seq with the differentially expressed genes obtained by RNA-Seq to screen out the genes in the differential tables located in the candidate regions. We used the National Center for Biotechnology Information database (www.ncbi.nlm.nih.gov, accessed on 23 August 2022) and combined it with GO and KEGG annotation results to further determine the functions of candidate genes and analyze the molecular mechanisms that affect the plant heights of foxtail millets.

### 4.7. Validation of Real-Time PCR

RT-qPCR was used to verify the reliability of transcriptome results and the final candidate genes. We took a 1 ug RNA sample from the RNA-Seq for RT-PCR. We used a PrimeScript^TM^ RT kit with gDNA Eraser (Takara, Dalian, China) for cDNA reverse transcription according to the instructions. The primers were designed using Primer 5 software with actin as the internal reference gene. SYBR^®^ Premix Ex TaqTM II (TliRNaseH Plus) was used to complete all RT-PCR analyses on the fluorescence quantitative PCR instrument (ABI7500, Applied Biosystems, Foster City, CA, USA). Each parent had three biological replicates and each biological replicate had three technical replicates; the relative expression levels of target genes were calculated by the 2^−ΔΔCt^ method.

## 5. Conclusions

The foxtail millet not only plays an important role in agriculture in arid areas but also plays an important role in C4 crop breeding. It is an important means to promote the green revolution to excavate the dwarf gene of the millet and explore the dwarf formation mechanism. In this study, 5 QTLs and 128 candidate genes related to the formation of the millet plant height were obtained by BSA-Seq. Through RNA-Seq, 2817 DEGs related to plant height were identified and a MAPK signaling pathway and metabolic pathway were determined to have important effects on the formation of the millet plant height. A joint analysis by BSA-Seq and RNA-Seq finally narrowed the QTLs further to two regions at the tip of chromosome IX and chromosome I, and nine candidate genes were identified. Finally, combined with previous studies, we proposed a hypothetical mechanism for the formation of foxtail millet dwarfing, and speculated that metabolism and MAPK signaling play important roles in the formation of the plant height in foxtail millet. Unfortunately, whether all the conclusions are correct still needs further experiments for verification. Therefore, we will verify the nine candidate genes and our hypothesis by transgenic means in future studies.

The comprehensive information presented here provides a reference for understanding the dwarfing mechanism of millet and provides some important genes related to dwarfing, which lays a foundation for the breeding of foxtail millet as a pioneer crop in arid areas and the study of the C_4_ model crop.

## Figures and Tables

**Figure 1 ijms-23-11824-f001:**
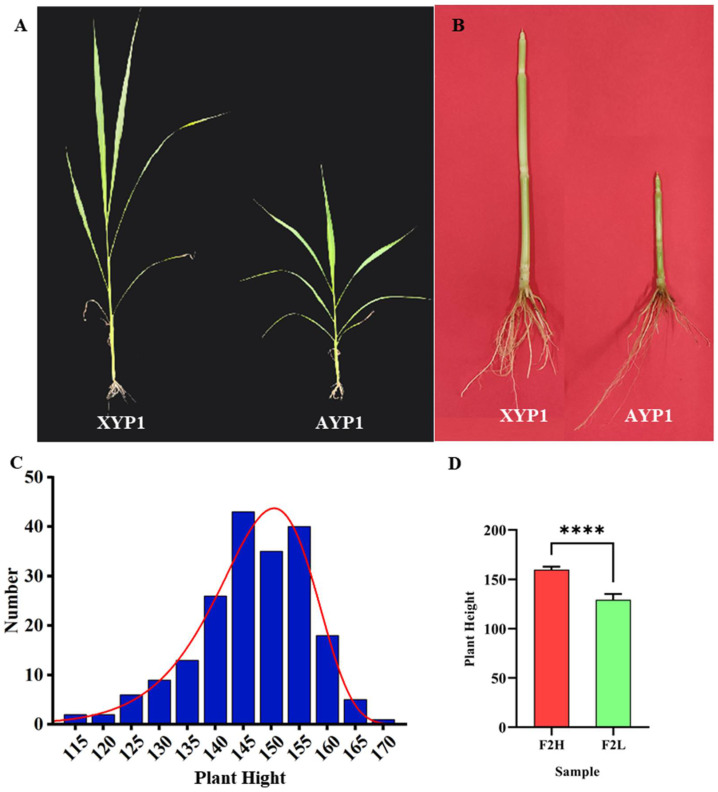
Field phenotype of foxtail millet’s plant height (**A**,**B**), plant height frequency distribution map of F_2_ population in the field (**C**), and the performance of plant height in two extreme mixed pools of plant heights (**D**). Asterisks indicate significant differences at various thresholds (**** *p* < 0.0001). Error bars represent the mean SE of 30 plants of foxtail millet in the mixed pool.

**Figure 2 ijms-23-11824-f002:**
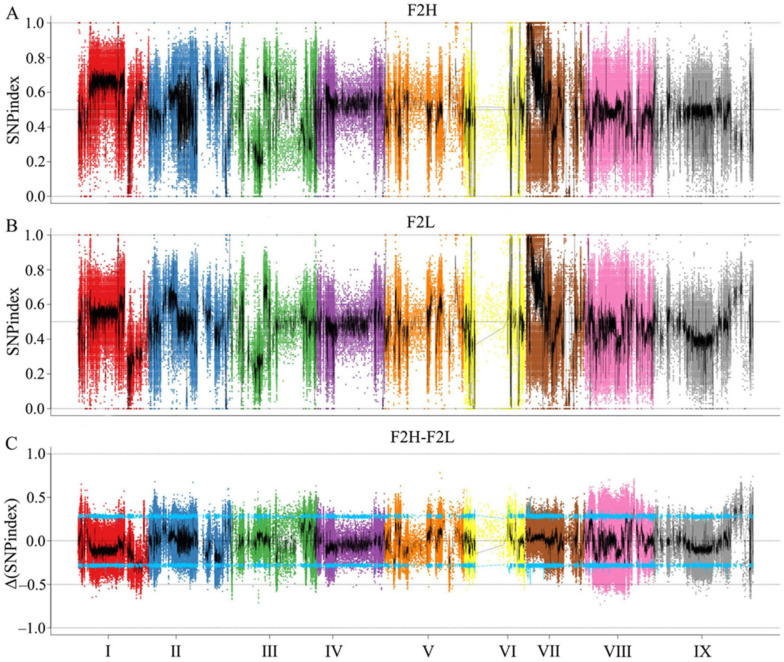
Distribution of SNP index association values on chromosomes. (**A**) Distribution of SNP- index values of F2H bulk on chromosomes. (**B**) Distribution of SNP index values of F2L bulk on chromosomes. (**C**) Distribution of Δ (SNP index) value on chromosomes, where the blue line represents the 95% CI.

**Figure 3 ijms-23-11824-f003:**
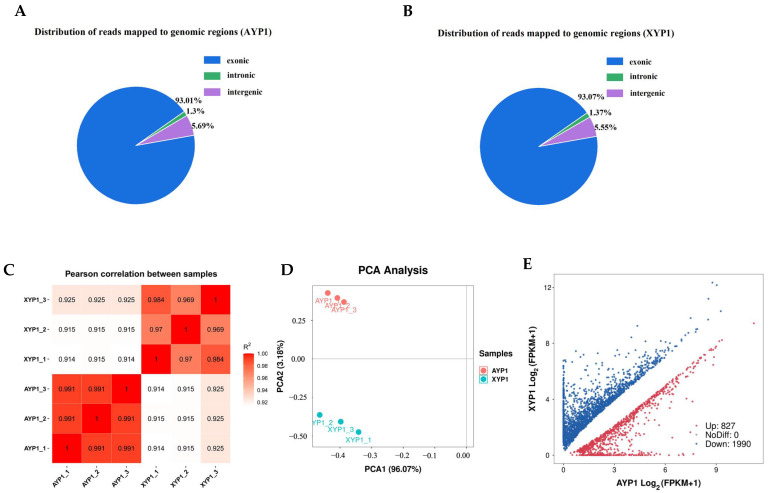
The results of reads mapping (**A**, **B**), the correlation analysis of six samples (**C**), the PCA analysis of six samples (**D**), and the differentially expressed genes between two parents (**E**).

**Figure 4 ijms-23-11824-f004:**
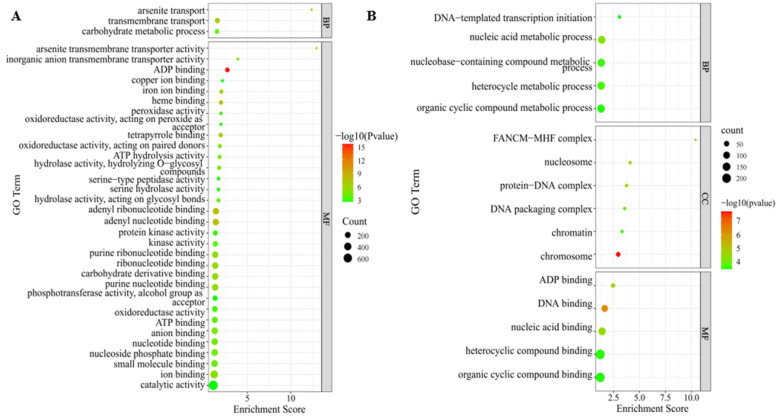
Gene Ontology (GO) enrichment analyses of DEGs obtained by RNA sequencing (RNA-Seq). Top 35 GO terms enriched for downregulated DEGs in dwarf samples (**A**), and top 15 GO terms enriched for upregulated DEGs in dwarf samples (**B**). Among them, different colors represent the size of the *p*-value, and the circular size represents the number of enriched. MF represents molecular function, BP represents biological processes, and CC represents cellular components.

**Figure 5 ijms-23-11824-f005:**
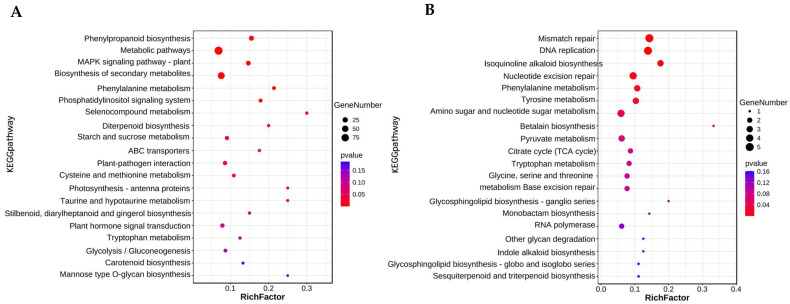
Top 20 KEGG pathways enriched by downregulated DEGs in dwarf samples (**A**), and top 20 KEGG pathway terms enriched by upregulated DEGs in dwarf samples (**B**). Among them, different colors represent the size of the *p*-value, and the circular size represents the number of enriched.

**Figure 6 ijms-23-11824-f006:**
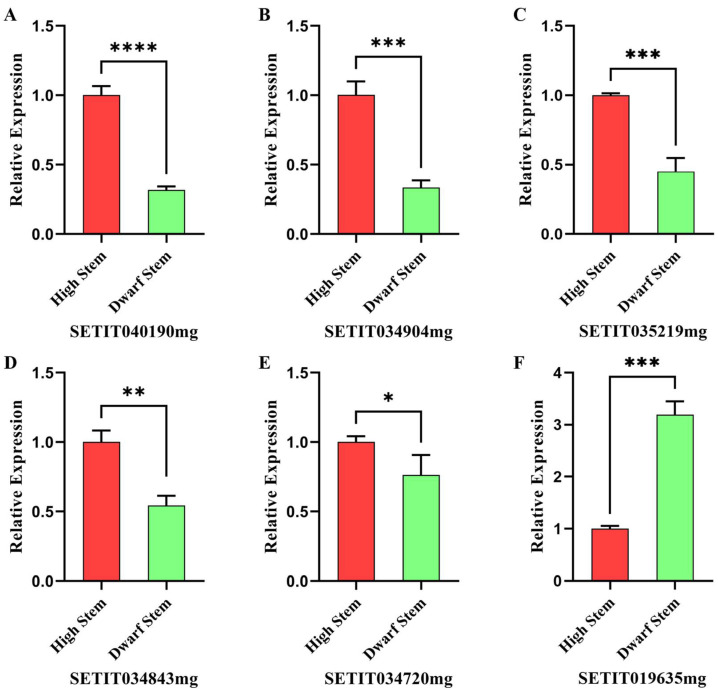
The results of quantitative RT-PCR (qRT-qPCR) validation. The actin gene was used as an internal control. The transcription level of the high-stem sample was set at 1.0. Asterisks indicate significant differences at various thresholds (* *p* < 0.05, ** *p* < 0.01, *** *p* < 0.001, **** *p* < 0.0001). Error bars represent the mean SE of three biological replicates. (**A**–**F**) Represent the relative expression of genes in different samples.

**Figure 7 ijms-23-11824-f007:**
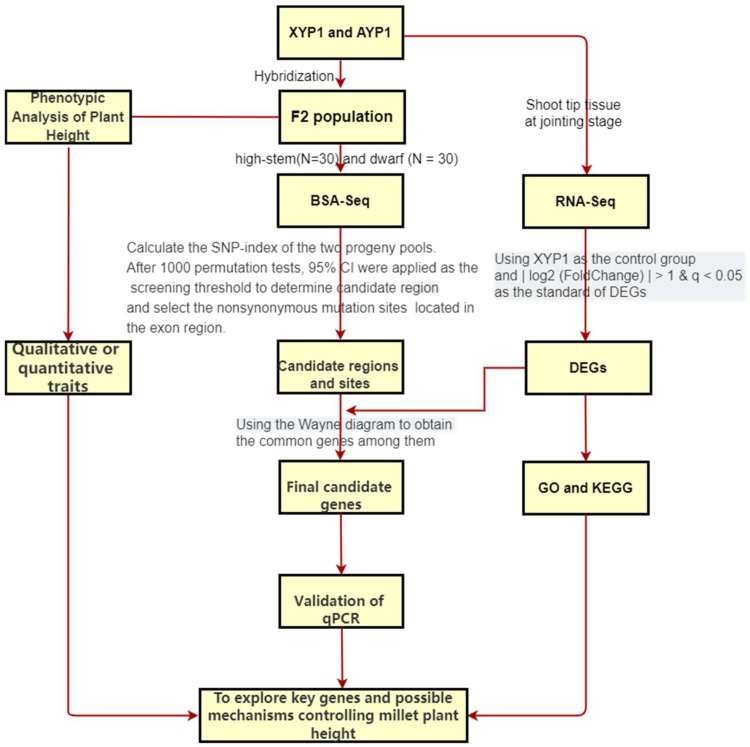
Technical route of this study.

**Table 1 ijms-23-11824-t001:** Quality statistics of mapping with the reference genome for BSA-Seq.

Sample	Mapped Reads	Total Reads	Mapping Rate (%)	Average Depth (×)	Coverage 1 × (%)	Coverage 4 × (%)
XAP1	36,115,445	36,886,738	97.91	10.57	95.49	90.82
AYP1	32,975,648	33,284,924	99.07	9.94	96.5	91.38
F2H	191,560,590	194,204,816	98.64	48.92	98.93	98.13
F2L	192,966,265	195,334,940	98.79	51.7	98.92	98.14

Mapped reads: the total number of reads on the reference genome was compared. Total reads: the total reads of valid sequencing data. Mapping rate: the number of reads on the reference genome was compared to the number of reads in the valid sequencing data. Average depth: the average sequencing depth, the total number of bases compared to the reference genome divided by genome size. Coverage 1×: the proportion of bases whose coverage depth is not less than 1× in the whole genome. Coverage 4×: the base coverage depth in the whole genome is no less than the base ratio of 4×.

**Table 2 ijms-23-11824-t002:** The statistics of mapping results for RNA-Seq.

Sample Name	Total Reads	Total Mapped Rate (%)	Multiple Mapped Rate (%)	Uniquely Mapped	Reads Map to ‘+’ (%)	Reads Map to ‘−’ (%)	Non-Splice Reads	Splice Reads
AYP1_1	51,123,404	94.75	3.18	91.57	45.78	45.78	55.73	35.84
AYP1_2	47,571,968	94.72	3.24	91.48	45.74	45.74	55.74	35.74
AYP1_3	45,527,652	94.89	3.24	91.66	45.83	45.83	55.91	35.74
XYP1_1	44,074,408	95.47	2.73	92.74	46.37	46.37	57.11	35.63
XYP1_2	55,221,554	94.92	3.09	91.83	45.91	45.91	55.89	35.93
XYP1_3	50,557,676	95.10	3.07	92.03	46.02	46.02	56.02	36.01

Total reads: the total reads after quality control. Total mapping rate: the rate of total reads that can be mapped to the reference sequence. Multiple mapping: the reads that are mapped to multiple positions in the reference sequence. Uniquely mapping: the reads that map to a unique position in the reference sequence. Reads mapping to +, reads mapping to –: the reads mapped to positive and negative chains, respectively. Non-splice reads: the reads that are mapped to only one exon. Splice reads: the same reads section mapped to different exons.

**Table 3 ijms-23-11824-t003:** Basic information of candidate genes.

Gene Name	Chromosome Location	Description
SETIT034720mg	IX: 57,492,650–57,497,221	Domain associated with HOX domains; homeobox protein BEL1 homolog
SETIT034843mg	IX: 51,772,544–51,7773,35	protein NRT1/PTR FAMILY 8.3
SETIT035219mg	IX: 58,779,433–58,782,708	Belongs to the major facilitator superfamily.
		Sugar transporter (TC 2. A. 1. 1) family
SETIT040190mg	IX: 51,799,242–51,801,341	protein NRT1/PTR FAMILY 8.3
SETIT033879mg	IX: 51,916,122–51,925,937	C-5 cytosine-specific DNA methylase
SETIT034904mg	IX: 57,439,447–57,442,618	Belongs to the sterol desaturase family; Fatty acid hydroxylase domain-containing protein
SETIT019635mg	I: 37,589,455–37,591,340	E2F/DP family winged-helix DNA-binding domain
SETIT017539mg	I: 36,542,422–36,544,052	Belongs to the peptidase A1 family
SETIT020559mg	I: 37,443,258–37,443,787	hypothetical proteins

## Data Availability

Not applicable.

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
