# Peer review of "Conuping BSA-Seq and RNA-Seq Reveal the Molecular Pathway and Genes Associated with the Plant Height of Foxtail Millet (Setaria italica)"

_ijms, 2022, doi:10.3390/ijms231911824_

Round 1

Reviewer 1 Report

In the manuscript  (ijms-1923742), the authors narrowed down candidate genes associated with dwarf phenotype in foxtail millet. A combination of BSA-Seq and RNA-Seq is likely effective in finding new candidate genes related to important phenotypes in foxtail millet and other plant species.

1. The authors selected 3 transporter genes from 9 candidate genes. However, this speculation was not supported clearly enough. I think that 9 candidate genes were enough and valuable findings. The authors should reveal the biological function of 3 genes in the future studies.

2. BSA-seq showed that 3 different candidate regions. But the combined analysis with RNA-Seq showed that no candidate genes were located on the chromosome VIII. The appropriate explanation should be provided. The authors focus the plant height between XYP1 and AYP1 in the manuscript. However, the differences of other phenotypes might be observed between XYP1 and AYP1.  

3. The authors mentioned that the 128 candidate genes were identified by BSA-Seq analysis. The chromosome location, accession number, and description of these genes is necessary to evaluate the results of BSA-Seq analysis.

4. The size of Figs. 4 and 5 were too small to understand them. In addition, the font in Figs. 3, 4 and 5 were too small.

5. Explanation for the selection strategy of candidate genes should be provided. The selection flowchart with gene number and criteria of statistical analysis likely support to understand the selection strategy.

6. The data of BSA-Seq and RNA-Seq data should be deposited to the public database.

Author Response

Response to Reviewer 1 Comments

(For more details, please refer to the attachment.)

Point 1: The authors selected 3 transporter genes from 9 candidate genes. However, this speculation was not supported clearly enough. I think that 9 candidate genes were enough and valuable findings. The authors should reveal the biological function of 3 genes in the future studies.

Response 1: Thank you very much for your valuable comments on the revision of our manuscript. Based on your opinions, we finally modified the description of Abstract and Conclusion as ”nine candidate genes related to plant height were obtained on chromosomes I and IX” (Line 21) and “and nine candidate genes were identified” (Line 634).

Point 2: BSA-seq showed that 3 different candidate regions. But the combined analysis with RNA-Seq showed that no candidate genes were located on the chromosome VIII. The appropriate explanation should be provided. The authors focus the plant height between XYP1 and AYP1 in the manuscript. However, the differences of other phenotypes might be observed between XYP1 and AYP1. 

Response 2: Thank you for pointing out the shortcomings of our manuscript. We studied the XYP1 and AYP1 with large difference in plant height to explore the formation mechanism of plant height. Of course, there were differences in other traits between XYP1 and AYP1 except plant height.

    However, the samples we used for RNA-Seq were the stem tip tissues during the jointing stage of foxtail millet. During this period, the stem tip plays a key role in the formation of plant height.

    Based on the period and tissue we sampled, as well as the expression of candidate genes located on chromosome VIII in RNA-Seq results was small or no significant difference, we excluded the possibility of candidate genes controlling plant height on chromosome VIII.

    In order to make others more understand and believe our inference, we will add this explanation in the section 2.4.1 (Line 256), and use the expression matrix of candidate genes located on chromosome VIII as a supplementary material (Supplementary Table 8) .

Point 3: The authors mentioned that the 128 candidate genes were identified by BSA-Seq analysis. The chromosome location, accession number, and description of these genes is necessary to evaluate the results of BSA-Seq analysis.

Response 3: Thank you for pointing out our shortcomings. Here we add the basic information of the 128 candidate genes as Supplemental Table 3 (Line 143), which includes the Ensembl Gene Id, the location of the chromosome and the description of its function. 

The size of Figs. 4 and 5 were too small to understand them. In addition, the font in Figs. 3, 4 and 5 were too small.

Response 4: Thank you for pointing out our shortcomings. We have adjusted the size and fonts of Figs.3, 4 and 5.

Point 5: Explanation for the selection strategy of candidate genes should be provided. The selection flowchart with gene number and criteria of statistical analysis likely support to understand the selection strategy.

Response 5: Thank you very much for your suggestion. In order to give readers a clearer understanding of the research content of this manuscript, we have sorted out the research content and drawn a selection flow chart that should include gene number and statistical analysis criteria (Line520-525).

Point 6: The data of BSA-Seq and RNA-Seq data should be deposited to the public database.

Response 6: Thanks for your reminder that we have uploaded the raw data from the trial to the NCBI. BioProject ID: PRJNA884512

Reviewer 2 Report

Plant height is an important agronomic trait related to yield and product quality. It is necessary to explore the key genes related to the dwarf phenomenon in foxtail millet. In this study, the BSA-Seq and RNA-Seq analysis were performed to explore the molecular pathway and genes associated with plant height of foxtail millet. The content of this paper has certain novelty. But there existed some problems, such as the analysis method, function study. The article needs major revision.

1. Result (2.4.1) I do not understand the analysis method, how the nine genes were isolated? There exists 128 candidate gene, only 9 nine key genes? (The 128 candi-247 date genes obtained by BSA-Seq were compared with 2817 differentially expressed genes 248 obtained by RNA-Seq after they were converted into a unified naming standard)

2. GA is an important hormone to regulate the plant height. There are no DEGs related to GA pathway?

3. Among 9 nine genes, which is major gene or minor gene? Whether these 9 genes can regulate the plant height? This conclusion needs functional verification, in model plant Arabidopsis or tobacco. Transgenic Foxtail millet is best.

Author Response

Response to Reviewer 2 Comments

(For more details, please refer to the attachment.)

Point 1: Result (2.4.1) I do not understand the analysis method, how the nine genes were isolated? There exists 128 candidate gene, only nine key genes? (The 128 candi date genes obtained by BSA-Seq were compared with 2817 differentially expressed genes obtained by RNA-Seq after they were converted into a unified naming standard).

Response 1: I am sorry that my unclear description has caused difficulties in your understanding of our analytical methods. In order to better carry out joint analysis, the reference genome used by BSA-Seq and RNA-Seq were the same, so that whether we obtain candidate genes from BSA-Seq or DEGs from RNA-Seq, the gene's naming is exactly the same. We call this the conversion of them into a unified naming standard. Then we compared the 128 candidate genes obtained by BSA-Seq with the 2817 DEGs obtained by RNA-Seq using the Wayne diagram to obtain the common genes among them, and finally obtained 9 candidate genes.

      In order to give readers a better understanding of my method, I modified my description in Result (2.4.1) as “We compared the 128 candidate genes obtained by BSA-Seq with the 2817 DEGs obtained by RNA-Seq using the Wayne diagram to obtain the common genes among them” (Line 246-249). At the same time, in order to better help readers understand our screening process and criteria, we drew a flow chart of screening candidate genes with explanations according to the recommendations of reviewer 1(Line 520-525).

Point 2: GA is an important hormone to regulate the plant height. There are no DEGs related to GA pathway? 

Response 2: Thank you for your prompt to us. However, we examined all the KEGG pathways enriched by our DEGs again and did not find that DEGs were enriched in the pathways related to GA. Therefore, we speculate genes that related to GA may not be the main genes causing plant height differences in this population. Of course, your doubts also provides ideas for our future research. In the next research, we will study the effect of GA on foxtail millet plant height as a key direction.

      At the same time, in this manuscript, in order to dispel doubts about our conclusions, we upload the results of KEGG enrichment analysis as supplementary materials with our manuscript (Line 218).

Point 3: Among 9 nine genes, which is major gene or minor gene? Whether these 9 genes can regulate the plant height? This conclusion needs functional verification, in model plant Arabidopsis or tobacco. Transgenic Foxtail millet is best.

Response 3: Thank you for your useful comments. We agree that if the relationship between the nine candidate genes and their effects on plant height are verified by transgenic methods, the credibility of this study and the quality of this manuscript will be greatly improved. However, transgenic experiments will be difficult to complete in a short time. Therefore, we will increase our description of the limitations of this experimental results in the Conclusion section of the manuscript (Line 637-640), and will use transgenic methods to verify these 9 candidate genes in future experiments. 

Reviewer 3 Report

Typing error in line 312.

Adjusted reference format in line 716, 753.

Author Response

Response to Reviewer 3 Comments

Point 1: Typing error in line 312.

Response 1: Thank you for your reminder. We have changed the wrong spelling of "he" to "The".

Point 2: Adjusted reference format in line 716, 753.

Response 2: Thanks for pointing out our mistake, we have deleted the reference of Line 716 because we think this reference is unnecessary. At the same time,we also corrected the reference file format of Line 753 as follows :

Ichimura, K.; Shinozaki, K.; Tena, G.; Sheen, J.; Henry, Y.; Champion, A.; Kreis, M.; Zhang, S.; Hirt, H.; Wilson, C., Mitogen-activated protein kinase cascades in plants: a new nomenclature. Trends in plant science 2002, 7, (7), 301-308. (Line 766-767)

     In addition, we have corrected five other error reference formats.

Round 2

Reviewer 2 Report

The authors answered the related questions, such as the analytic method,  partial results, and supplemented the related informatin. I recommed that the article can be accepted.